**RESEARCH**

# Genomic diversity and ecology of human-associated *Akkermansia* species in the gut microbiome revealed by extensive metagenomic assembly

Nicolai Karcher[1†], Eleonora Nigro[2†], Michal Punčochář[1], Aitor Blanco-Míguez[1], Matteo Ciciani[1], Paolo Manghi[1], Moreno Zolfo[1], Fabio Cumbo[1], Serena Manara[1], Davide Golzato[1], Anna Cereseto[1], Manimozhiyan Arumugam[2], Thi Phuong Nam Bui[3], Hanne L. P. Tytgat[3,4], Mireia Valles-Colomer[1†], Willem M. de Vos[3,5†] and Nicola Segata[1,6*†]

* Correspondence: nicola.segata@unitn.it
†Nicolai Karcher and Eleonora Nigro contributed equally to this work.
†Mireia Valles-Colomer, Willem M. de Vos, and Nicola Segata are co-senior authors.
¹Department CIBIO, University of Trento, Trento, Italy
⁶IEO, European Institute of Oncology IRCCS, Milan, Italy
Full list of author information is available at the end of the article

## Abstract

**Background:** *Akkermansia muciniphila* is a human gut microbe with a key role in the physiology of the intestinal mucus layer and reported associations with decreased body mass and increased gut barrier function and health. Despite its biomedical relevance, the genomic diversity of *A. muciniphila* remains understudied and that of closely related species, except for *A. glycaniphila*, unexplored.

**Results:** We present a large-scale population genomics analysis of the *Akkermansia* genus using 188 isolate genomes and 2226 genomes assembled from 18,600 metagenomes from humans and other animals. While we do not detect *A. glycaniphila*, the *Akkermansia* strains in the human gut can be grouped into five distinct candidate species, including *A. muciniphila*, that show remarkable whole-genome divergence despite surprisingly similar 16S rRNA gene sequences. These candidate species are likely human-specific, as they are detected in mice and non-human primates almost exclusively when kept in captivity. In humans, *Akkermansia* candidate species display ecological co-exclusion, diversified functional capabilities, and distinct patterns of associations with host body mass. Analysis of CRISPR-Cas loci reveals new variants and spacers targeting newly discovered putative bacteriophages. Remarkably, we observe an increased relative abundance of *Akkermansia* when cognate predicted bacteriophages are present, suggesting ecological interactions. *A. muciniphila* further exhibits subspecies-level genetic stratification with associated functional differences such as a putative exo/lipopolysaccharide operon.

**Conclusions:** We uncover a large phylogenetic and functional diversity of the *Akkermansia* genus in humans. This variability should be considered in the ongoing experimental and metagenomic efforts to characterize the health-associated properties of *A. muciniphila* and related bacteria.

## Introduction

The human body is home to several distinct microbiomes which represent functionally and phylogenetically diverse microbial ecosystems that are key for human health [1–3]. A frequent and abundant inhabitant of the gut microbiome is *Akkermansia muciniphila*, a Gram-negative, non-motile anaerobic bacterium specialized in the degradation of mucin [4]. *A. muciniphila* can utilize mucin as its sole carbon and nitrogen source [4]; thus, growth in its natural habitat is not directly dependent on the influx of dietary compounds. *A. muciniphila* continues to attract attention due to its association with host health: the relative abundance of *A. muciniphila* is inversely correlated with obesity in humans [5, 6], and it was shown to alleviate insulin resistance and obesity while increasing gut barrier function in a mouse model of diet-induced obesity [7]. Its potential as a next-generation probiotic in the battle against metabolic disorders was confirmed in a first intervention trial targeting humans with metabolic syndrome and obesity [8].

The human microbiome hosts a vast bacterial diversity at the level of distinct strains belonging to the same species (i.e., conspecific strains) [3, 9–13]. The genomic variation of conspecific strains often exceeds 3% nucleotide variation in the core genes, and when comparing pairs of conspecific strains, it is frequently observed that 25% of genes are present in only one of the two, causing each human microbiome to be unique at the strain level [3]. Importantly, this subspecies genomic variability translates into phenotypic variability, for example, in connection with host lifestyle [14–16] and at the immunological level [17, 18]. However, experimental *Akkermansia* research still heavily relies on the type strain *A. muciniphila* Muc$^T$ (ATCC BAA-835) and on a few more genomes of newly isolated strains that became available recently [19–21]. Furthermore, only a single other species in the *Akkermansia* genus—*A. glycaniphila* (Pyt$^T$, DSM 100705)—has so far been described and genomically characterized [22, 23]. There is thus the urgent need to expand our understanding of the genomic variation and (sub)-species diversity of *Akkermansia* for improving both the interpretation of its functions and its potential use in biomedicine.

Recently, a large number of publicly available metagenomes of human-associated microbial communities have been mined to produce hundreds of thousands of metagenome-assembled genomes (MAGs) [3, 24–26] and methods to profile and investigate strains directly in metagenomes have become increasingly effective [9, 10, 27–29]. While these tools offer the opportunity to characterize population genomics of important but poorly characterized human-associated bacteria, only a few species have been investigated so far at high genomic resolution and global scale [30–33].

Here, we present a comprehensive genomic characterization of *Akkermansia muciniphila* and closely related *Akkermansia* spp., using a total of 2226 MAGs belonging to the *Akkermansia* genus, 188 publicly available isolate genomes, and 6 newly sequenced isolate genomes. The *Akkermansia* MAGs were obtained by expanding our recent catalog of human-associated MAGs [3] with 166,518 additional MAGs from 45 different datasets comprising samples also from mice and non-human primates for an integrated catalog of 321,241 MAGs (see the "Methods" section). Next to the species-level clade with the *A. muciniphila* type strain, we show the existence of four other *Akkermansia* candidate species that colonize the human gut. These five candidate species display strong co-exclusion within a given host, are phylogenetically stratified at the subspecies

level, and are at the same time widely distributed across hosts, age, and geography. Comparison of candidate species shows differential association with BMI in humans and vitamin B12 synthesis capabilities. We also analyzed the genomic organization of CRISPR-Cas loci (providing adaptive immunity against foreign DNA [34]) in *Akkermansia* candidate species and found these to differ in their locus architecture and spacer numbers. We furthermore identified de novo assembled putative bacteriophages with spacer hits from *Akkermansia* candidate species and found that viral detectability correlates strongly with the relative abundance of cognate *Akkermansia* candidate species, suggesting an intimate ecological interplay. These, and other genomic analyses in this work, provide a solid basis for future mechanistic explorations and biomedical applications of *Akkermansia*.

## Results and discussion

### A large-scale metagenomics-based analysis of *Akkermansia* candidate species

In order to study the diversity of *Akkermansia* species in the human microbiome, we collected all genomes available from isolate sequencing as well as MAGs from large collections of metagenomic datasets and unified them into a single genomic resource. Specifically, we gathered and quality controlled 119 isolate genomes from NCBI that were taxonomically annotated as *A. muciniphila*, as well as 69 labeled as *Akkermansia* sp. [4, 5, 19, 22, 35–41]. We further obtained 2226 MAGs taxonomically annotated to the genus *Akkermansia* from a total of 18,600 shotgun metagenomes (see the "Methods" section) sampled from multiple hosts including humans, non-human primates, and mice. Only high-quality MAGs—defined as those with at least 90% estimated genomic completeness and at most 5% estimated genomic contamination [42]—were included in the analysis. We further enhanced our genome set with 6 isolate genomes [43]. The integrated *Akkermansia* genome resource we consider for downstream analysis thus consists of a total of 2420 genomes (Additional file 1: Table S1).

### Multiple under-characterized *Akkermansia* candidate species are present in the human microbiome

We reconstructed the phylogeny of all genomes in our set using the 400 universal marker genes adopted in PhyloPhlAn 3 [44–46] (Fig. 1A, the "Methods" section) including *Verrucomicrobium spinosum* as an outgroup [4]. This revealed the presence of several well-defined monophyletic clades (Fig. 1A). In addition to the previously described *A. glycaniphila* species [22], these clades—following the validated species-level genome bins (SGBs) approach based on whole-genome genetic distances [3] (see the "Methods" section)—delineate candidate species (Fig. 1B). The candidate species are genetically distinct, with inter-candidate species genome-wide average estimated nucleotide identities generally below 90% (except between a single pair of candidate species, Fig. 1B). We confirmed those results using genome similarity estimates obtained using PhyloPhlAn 3 markers (Additional file 2: Figure S1). One of the five delineated candidate species (henceforth "*A. muciniphila*") includes the type strain of *A. muciniphila* (Muc$^T$) [4] as well as 108 isolate genomes. The remaining four candidate species (SGB9223, SGB9224, SGB9227, and SGB9228) comprise not only MAGs but also isolate genomes that were, however, taxonomically described as *A. muciniphila* or

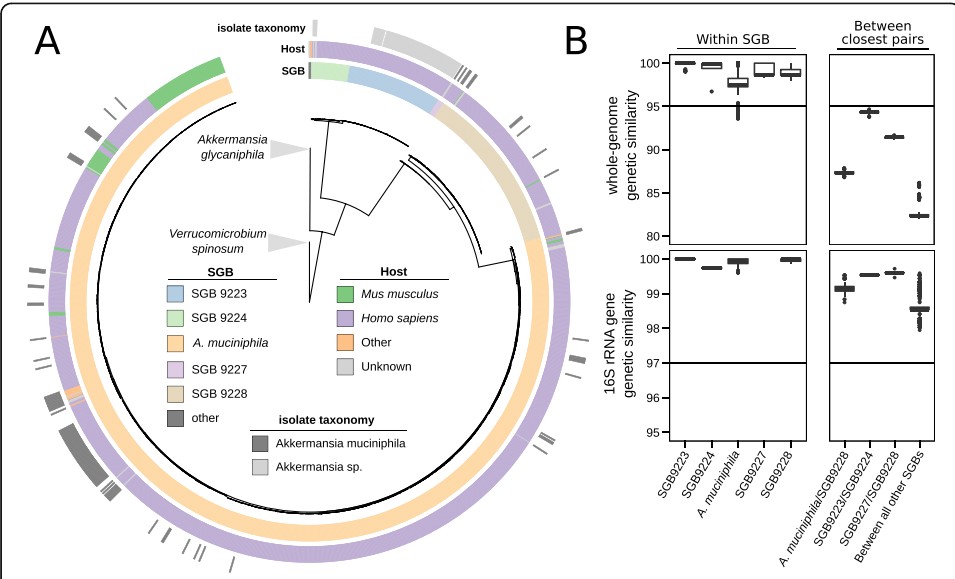

**Fig. 1** The *Akkermansia* genus comprises four additional candidate species phylogenetically rooted between the already characterized *A. glyceniphila* and *A. muciniphila*. **A** Whole-genome phylogeny of the 2420 metagenome-assembled genomes (MAGs) reconstructed here and the genomes from isolate sequencing available in NCBI taxonomically annotated as *A. muciniphila* or *Akkermansia* spp. The phylogenetic tree is rooted using *Verrucomicrobium spinosum* as an outgroup and was built using PhyloPhlAn 3 [46] with 400 universal markers (see the "Methods" section). SGB, species-level genome bin (see the "Methods" section). **B** Within- and between-clade whole-genome average estimated nucleotide identity (fastANI [47], top panels) and full-length 16S sequence distances (bottom panels) among *Akkermansia* SGBs provide evidence that these are candidate species

*Akkermansia* spp. in NCBI. Cultivated members of the candidate species were retrieved not only from humans, but also from mice, non-human primates, and—very rarely— other mammals, such as elephants, horses, and reindeers [20]. *A. glyceniphila* was originally isolated from a python [22], and we did not uncover new diversity for this species in the available datasets, suggesting that *A. glyceniphila* is not found in mammals. A reason for the taxonomic assignment of cultivated strains to the *A. muciniphila* species and the generally underestimated diversity of the genus is probably the surprisingly high similarity displayed by 16S rRNA gene sequences of these strains, with 16S rRNA gene sequences of strains in different candidate species never diverging by more than 2% (Fig. 1B, the "Methods" section). Taken together, these data show that a total of five *Akkermansia* candidate species exist in the human, mouse, and non-human primate gut microbiomes, four of them remaining under-investigated and uncharacterized.

### *Akkermansia* candidate species are enriched in humans and co-exclude within a host

We next set out to assess host specificity, co-abundance patterns, and metadata associations of the *Akkermansia* candidate species. To this end, we first developed a marker-based method with increased sensitivity compared to metagenomic assembly to detect the presence and relative abundance of *Akkermansia* candidate species in metagenomes. In brief, this was done by (1) identifying genes that were core to each of the *Akkermansia* candidate species and at the same time never detected in other *Akkermansia* or non-*Akkermansia* species-level groups (*marker genes*) and (2) using these

marker genes as targets for read-mapping inside MetaPhlAn 3.0 [48, 49] to estimate their coverage and relative abundance (see the "Methods" section). By profiling the 13, 237 metagenomic samples from 98 publicly available datasets with sufficient metadata (Additional file 1: Table S2), we found that *Akkermansia* candidate species differed strongly in their prevalence across hosts. *A. muciniphila* is by far the most prevalent candidate species across all hosts, being detected in 34% of adult humans and reaching a maximum prevalence of 54% in laboratory-held mice (Fig. 2A). The other candidate species were detected at lower prevalence (< 25%) across all hosts (Fig. 2A). Interestingly, *Akkermansia* candidate species were found to be much more often present in captive animals than in free-living ones: while laboratory mice and mice humanized via microbiome transplantation are fairly likely to host *A. muciniphila*, SGB9224, or

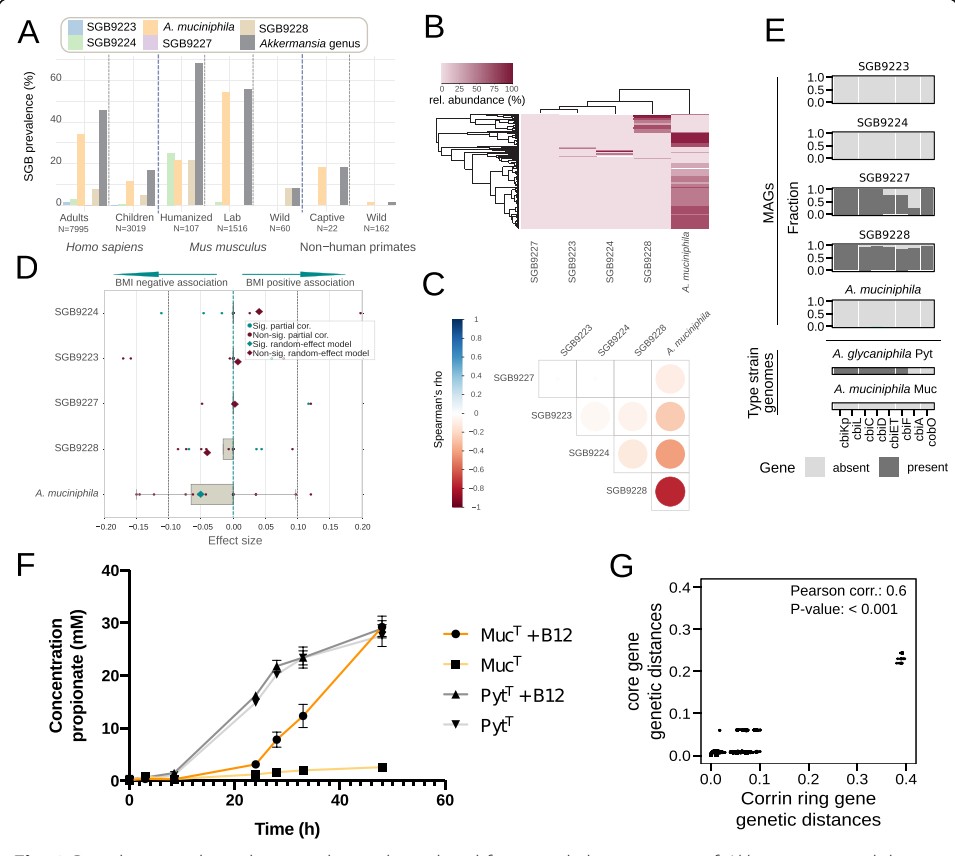

**Fig. 2** Prevalence and insights into the ecological and functional characteristics of *Akkermansia* candidate species. **A** *Akkermansia* candidate species have variable prevalence across hosts and wild versus captive mice and non-human primates. We computed prevalences using species-specific marker genes (see the "Methods" section) applied on a total of 13,237 metagenomic samples. **B**, **C** *Akkermansia* candidate species are strongly mutually exclusive (analysis based on 4171 *Akkermansia*-positive human metagenomes). **D** *A. muciniphila* but not the other *Akkermansia* candidate species is associated with decreased host body mass index (BMI) according to a meta-analysis random effect model of partial correlations adjusted for age and sex (see the "Methods" section) comprising 3311 human metagenomic samples from 22 datasets (Additional file 1: Table S2). **E** Corrin ring biosynthesis operon genes are consistently present only in candidate species SGB9227 and SGB9228 (see the "Methods" section). **F** Growth analysis of the *A. muciniphila* and *A. glycaniphila* type strains shows propionate production by Pyt[T] but not Muc[T] in the absence of vitamin B12. This is indicative of endogenous production of vitamin B12 (acting as a cofactor for the methyl-malonyl CoA synthase reaction) by Pyt[T] but not Muc[T]. **G** Core gene genetic distances are correlated with corrin ring biosynthesis gene genetic distances. Pairwise distances were computed only for strains in which all genes were found together on the same contig

SGB9228 (up to 54% prevalence), in wild mice, solely SGB9228 was detected in only 5 mice from a single study (out of 60 mice from 3 different studies being assessed; 8% prevalence). Similarly, merely two out of 162 samples from wild non-human primates tested positive for any *Akkermansia* candidate species (1.2% prevalence). Despite potential biases due to uneven sampling and effects of diet, these data suggest a marked specificity of *Akkermansia* candidate species for the human gut (with the exception of *A. glycaniphila*), and while strains from these candidate species can colonize mice and non-human primates, such colonization appears to be predominantly a consequence of man-made environments, suggesting colonization from care-taking humans as a plausible mechanism.

While *Akkermansia* candidate species are found in almost half of all human samples, the presence of one is strongly anti-correlated with the others (Fig. 2B, C): it is extremely rare to detect more than one candidate species present in the same host, with only 46 instances of two co-occurring candidate species in human metagenomes out of 4171 cases in which at least one was present (corresponding to a co-occurrence rate of ~1%), and no instance of more than two co-occurring candidate species. These five closely related candidate species thus show a mutual exclusion pattern suggestive of complex and possibly host-mediated ecological interactions that remain to be explored.

### *A. muciniphila* but not the other candidate species is associated with lower BMI

The presence and abundance of *A. muciniphila* in the gut microbiome has been negatively associated with body mass index in previous studies based on 16S rRNA gene sequencing [50, 51], and the link was shown to be causal in both mice and humans by supplementation with cells of *A. muciniphila* Muc$^T$ [7, 8]. Because of the limitations of 16S rRNA gene amplicon sequencing in distinguishing *Akkermansia* candidate species (Fig. 1B), we performed an analysis on the association between relative abundances of individual *Akkermansia* candidate species and BMI. We considered 3311 samples in 22 different metagenomic datasets from five continents and adjusted for age and sex in a random effect model meta-analysis (Additional file 1: Table S3). Interestingly, only the relative abundances of *A. muciniphila* were found to be significantly negatively associated with BMI, while associations of other candidate species were not statistically significant (Fig. 2D), suggesting that *A. muciniphila* should be regarded as the primary candidate for microbiota-based therapeutic interventions aimed at improving host metabolic health (as a recent proof-of-concept trial also reported [8]).

We next tested whether available host characteristics other than BMI were associated with *Akkermansia* candidate species relative abundances and also examined whether genetic stratification by host parameters could be detected within candidate species. At the candidate species-level, no association with age was detected, while sex (as self-declared by the individuals) was associated with the relative abundance of *A. muciniphila* (after adjusting for age and BMI), with women harboring comparatively higher relative abundances (*P*-value = 4.8e−05), as observed elsewhere [52]. To test for associations of candidate species with host metadata at the level of their internal phylogenomic structure, we subsequently computed PERMANOVA statistics for all combinations of single candidate species and host, age, geography, and Westernization status. While some significant associations were identified (especially for *A. muciniphila*

and SGB9228), the largest effect size among the significant ($P$-value $< 0.05$) tests was a PERMANOVA $R^2$ of 0.10 for SGB9228 with continent (Additional file 2: Figures S2, S3, S4), suggesting that no strong associations of strain-level structure with host meta-data are detectable.

With *Akkermansia* supplementation becoming available [8, 53], it is relevant to verify that such interventions are not potentially causing microbial anti-drug resistances to spread in the human gut. To this end, we first screened all *Akkermansia* genomes for antibiotic resistance genes using the Comprehensive Antibiotic Resistance Database (CARD) [54]. Overall, we found only 4 genes known to be involved in antibiotic resistance present in more than 1% of all genomes. Among those, only *adeF* (encoding a membrane protein of a drug efflux complex) is consistently found in most genomes (prevalence of 81% over all genomes), but still never present in SGB9223, SGB9224, nor SGB9227 (Additional file 2: Figure S5). In addition to these well-cataloged resistance genes, a recent study reported the presence of 8 genes (including 3 antibiotic resistance genes) in *A. muciniphila* strain GP36 derived from the broad-range plasmid RSF1010 that is found in many gram-negative bacteria including *E. coli* [19]. We queried all genomes for the presence of this plasmid-derived sequence and found 55 genomes (2.2% overall prevalence) in which we could detect at least 50% of the sequence of RSF1010 at 70% average sequence identity or higher. A total of 49 of the 55 instances were found in *A. muciniphila* (2.5% prevalence in *A. muciniphila*). In all 55 positive cases, these genes were found on contigs larger than the plasmid (~8 kb), suggesting that they may be integrated into the bacterial genome (as also reported in [19]). Of note, the *A. muciniphila* type strain Muc$^T$ carries no antibiotic resistance genes and its use does not raise any antibiotic resistance concern as also indirectly confirmed by dose scaling pilot studies in humans and toxicological studies in rabbit and other model organisms [8, 55]; however, ongoing and future human trials with strains different from the type strain should carefully consider their antibiotic resistance potential. In conclusion, although the rare occurrence of antibiotic resistance genes from plasmid RSF1010 in some *A. muciniphila* genomes has evident safety implications for their use in therapeutic applications, our findings indicate that *Akkermansia* candidate species mostly lack genetic means to defend themselves against currently used antibiotics.

### Vitamin B12 synthesis capabilities were independently lost by two *Akkermansia* candidate species

Due to its essential nature and limited availability in the human gut, vitamin B12 (cobalamin) is regarded as a key element in host-microbe interactions [56]. In a recent study, 75 *Akkermansia* strains were reported to differ in their potential to produce this important cofactor [21]. We set out to characterize the vitamin B12 synthesis capabilities of the *Akkermansia* candidate species as well as *A. glycaniphila*. By identifying corrin ring biosynthesis genes as a proxy for vitamin B12 synthesis capability [56], we confirm that the large majority of MAGs from candidate species SGB9227 and SGB9228 encode most proteins involved in producing vitamin B12 (75% of SGB9227 MAGs encode all proteins except CbiA; 92% of SGB9928 MAGs encode all proteins except CbiF), while those genes were never found in *A. muciniphila*, SGB9223, nor SGB9224 (Fig. 2E). Interestingly, the more phylogenetically distant *A. glycaniphila* Pyt$^T$

[22] also contains 6 out of 8 corrin ring biosynthesis genes (Fig. 2E). The differential vitamin B12 synthesis capabilities of *Akkermansia* spp. were successfully validated by in vitro assays: propionate production (a proxy for vitamin B12 production, as the pathway includes the B12-dependent methyl-malonyl CoA synthase reaction [57]) was detected when growing *A. glycaniphila* Pyt$^T$ but not *A. muciniphila* Muc$^T$ in the absence of vitamin B12 (Fig. 2F, Additional file 2: Figure S6). The *cbiA* gene that we did not detect in the majority of SGB9227 MAGs is not found in *A. glycaniphila* Pyt$^T$ either, suggesting that this gene may not be necessary for B12 production in *Akkermansia* spp. Furthermore, we detected a strong correlation between pairwise genetic distances of corrin ring biosynthesis genes and core genes between strains of SGB9227, SGB9228, and the singular *A. glycaniphila* genome (Spearman rho = 0.6, *P*-value <0.001; Fig. 2G), suggesting that the B12 biosynthesis genes are ancestral to all *Akkermansia* candidate species and were lost by *A. muciniphila*, SGB9223, and SGB9224 candidate species in the human gut. The most likely evolutionary scenario would consist of two independent loss events: one after the most recent common ancestor of SGB9223/SGB9224 separated from the ancestor of the remaining candidate species, and another after the ancestor *A. muciniphila* separated from the ancestor of SGB9228 (Additional file 2: Figure S7). Taken together, these results reveal two independent B12 synthesis loss events in *Akkermansia* candidate species and indicate that new *Akkermansia* strains should be studied for their potential to increase colonic vitamin B12 biosynthesis.

### *Akkermansia* candidate species encode a novel variant of type I-C CRISPR-Cas loci

CRISPR-Cas systems are widely used by prokaryotes to fend off foreign DNA [58] and can be exploited to alter the microbiome makeup [59]. However, they have only been studied in detail for a limited number of bacteria, and strain-level variations have been documented [59]. We thus screened our catalog of *Akkermansia* genomes and MAGs for the presence of CRISPR-Cas loci. A great majority of genomes (68%, Fig. 3A) harbored at least one CRISPR-Cas locus, and while type I-C loci [61] were detected in all *Akkermansia* candidate species, *A. muciniphila* is the only species in the genus sometimes carrying a type II-C locus (33%, Fig. 3A). In 9% of the cases, *A. muciniphila* strains carried both the type II-C locus and the type I-C locus (Fig. 3A).

The structure of type I-C loci in *Akkermansia* candidate species differs notably from the canonical organization [61]: Cas3, Cas5, Cas8c, and Cas7 genes are encoded in the opposite direction of Cas4, Cas1, and Cas2, thus representing a novel variant of type I-C CRISPR-Cas loci. The majority of loci (62.4%) contain two CRISPR arrays, one upstream and one downstream of the Cas gene cassette. In contrast, the type II-C loci of *A. muciniphila* have the canonical structure [62] in 95% of the strains in which the locus was detected (Fig. 3B). The presence of a type II-C locus in *A. muciniphila* has no clear phylogenetic structure (Fig. 3C), highlighting a peculiar evolutionary history. *Akkermansia* candidate species also differ in the total number of spacer sequences encoded in each genome as well as the fraction of loci that contain two (instead of one) CRISPR arrays (Fig. 3B, D). SGB9223 and SGB9228 on average contain more spacer sequences (median 43 s.d. 15.4 and median 36 s.d. 29.5) compared to SGB9224, which has the lowest number of spacers (median 3, s.d. 16.4). Similarly, 91% of all genomes from SGB9223 contain two CRISPR arrays (one upstream, one downstream of the Cas

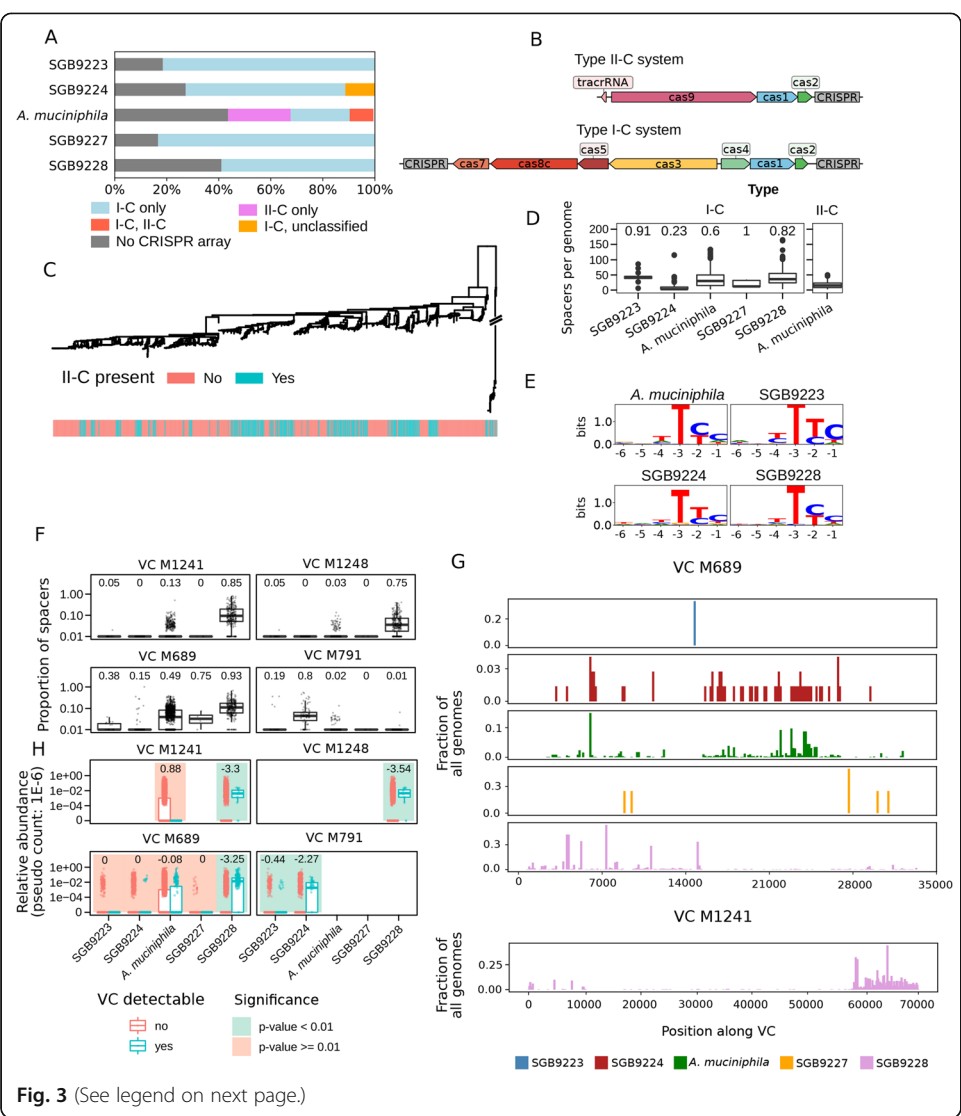

**Fig. 3** (See legend on next page.)

(See figure on previous page.)

**Fig. 3** The CRISPR-Cas system of *Akkermansia* candidate species and their viral targets. **A** CRISPR locus type composition of *Akkermansia* candidate species. All candidate species possess CRISPR locus type I-C, with the exception of *A. muciniphila* in which type II-C is present in more than 30% of the genomes. **B** Representative locus organization of CRISPR loci over *Akkermansia* candidate species. Some type I-C loci contain only one CRISPR array. Gene and CRISPR array lengths are scaled to correspond to the median length over all loci. **C** Phylogenetic tree of *A. muciniphila* subspecies colored by type II-C presence. **D** The total number of spacer sequences for the genomes in each *Akkermansia* candidate species. Type II-C loci were only found in *A. muciniphila*. Numbers above the boxplots correspond to the fraction of type I-C loci with two CRISPR arrays. **E** Logo plots of predicted PAM sequences in putative (phage) Viral Clusters (VCs, see the "Methods" section) upstream of sequences with perfect matches against CRISPR spacer sequences from type I-C loci. **F** Proportion of CRISPR spacers within candidate species genomes with a near-perfect match (at most 2 mismatched nucleotides) for four VCs. The number above the box plots corresponds to the fraction of genomes with at least one spacer hit against a given VC (see the "Methods" section). **G** Mapping of spacers from *Akkermansia* genomes against two representative VCs, visualized with a sliding window of 150 nt. See Additional file 2: Figure S8 for the remaining VCs. **H** Distribution of the relative abundances of the *Akkermansia* candidate species based on the presence or absence of each cognate VC in the metagenome (Additional file 1: Table S2, see the "Methods" section). *P*-values for differential abundance were determined via two-sided Wilcoxon rank-sum tests. *P*-values of <0.01 were considered significant. The numbers above the box plots correspond to the generalized fold change, with negative numbers indicating a higher bacterial abundance when a VC is detected [60]

gene cassette), whereas only 23% of genomes in SGB9224 do so (Fig. 3D). *Akkermansia* candidate species thus generally contain CRISPR-Cas systems, with distinct loci structure and spacer composition, indicating considerable divergence in their exposure to foreign DNA over their evolutionary trajectory.

## Newly discovered putative phages are recognized by *Akkermansia* CRISPR-Cas systems and tend to co-occur with cognate candidate species

We next identified de novo assembled, putative intestinal bacteriophages in shotgun gut viromes defining Viral Clusters (VCs, see the "Methods" section) and screened them for the presence of *Akkermansia* CRISPR-Cas spacers. We found no spacer hits against any of the known intestinal phages currently in RefSeq [63], but we instead detected a total of eight VCs with spacer hits (Additional file 2: Figure S8), four of which consistently attracted spacer sequences from at least one *Akkermansia* candidate species (Fig. 3F, see the "Methods" section) which we considered for further analysis. While some VCs exhibited hits from spacers from only one of the candidate species (SGB9228 for M1241 or M1248), other VCs (M689) were found to attract spacer sequences from all candidate species. Regardless of VCs, SGB9228 genomes on average have the highest total fraction of spacer sequences hit (Additional file 2: Figure S9).

The mapping of *Akkermansia* spacer sequences against VCs revealed that spacer sequences tend to cluster locally in the phage genome and that different locations on the viral genome attract spacers in a species-dependent fashion (Fig. 3G, Additional file 2: Figure S8). Furthermore, identification of the sequences directly upstream of all spacer sequence hits allowed reconstruction of the canonical type I-C protospacer adjacent motif (PAM) "TTC" (Fig. 3E) found in *Bacillus halodurans* [64]. The presence of multiple, distinct hits for some species-VC combination (Fig. 3G, Additional file 2: Figure S8) suggests that these matches are not spurious and that many combinations of *Akkermansia* candidate species and viral clusters reflect multiple bacterium-phage interactions in the intestinal environment. To further investigate potential ecological

interactions between *Akkermansia* candidate species and phages, we assessed the co-occurrence between candidate species and the matching VCs across 13,237 metagenomes (see the "Methods" section). For 5 out of 10 putatively interacting pairs of VCs and candidate species (defined as those pairs where more than 10% of genomes of the candidate species have at least one VC-matching spacer), we found that a candidate species is significantly more abundant (*P*-value <0.01) whenever the cognate VC is detectable (Fig. 3F, H, see the "Methods" section). Taken together, our analysis showed that CRISPR spacer sequences found in the genome of *Akkermansia* candidate species can be frequently mapped to four putative phages that co-occur with their cognate candidate species, suggesting that they are ecologically interacting in the human gut.

## *A. muciniphila* is stratified in four subspecies with different host preferences and functional profiles

In all bacterial species, a large fraction of the phenotypic variability is encoded at the subspecies level [5, 14–16]. We thus further focused on the intra-species genetic variation of *A. muciniphila* given its prevalence and relevance due to its association with lower host BMI (Fig. 2D). We found *A. muciniphila* to have four monophyletic subclades that we labeled Amuc1 to Amuc4 (Fig. 4A). We left strains unassigned that are not part of one of these monophyletic subclades (accounting for 29% of all *A.*

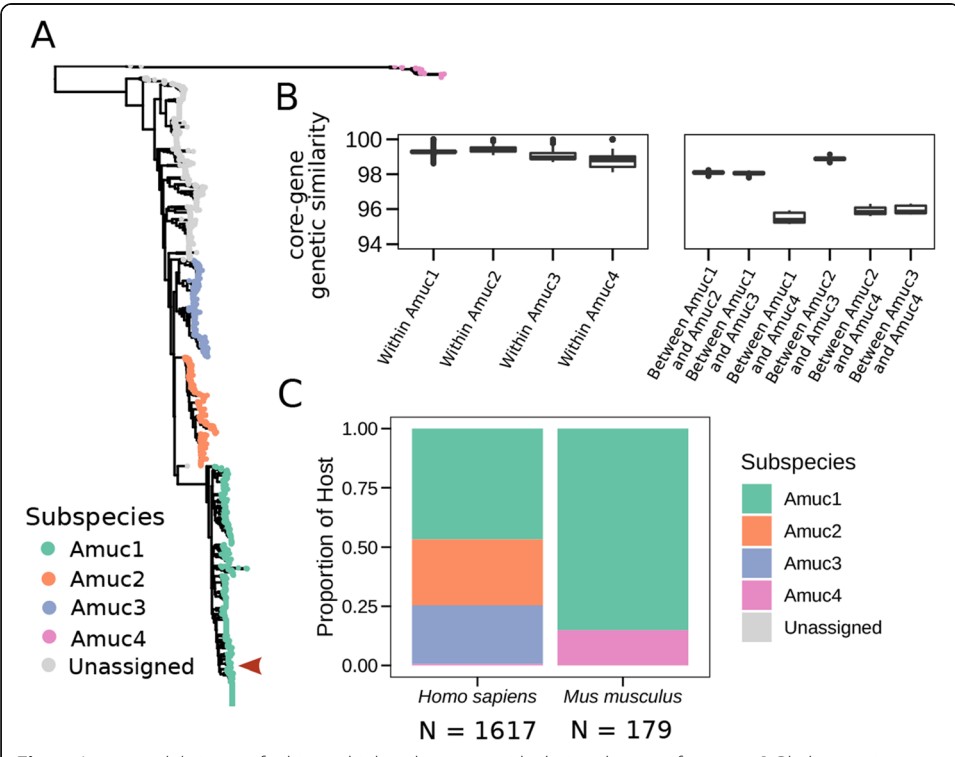

**Fig. 4** *A. muciniphila* is stratified in multiple subspecies with distinct host preferences. **A** Phylogenetic tree of *A. muciniphila* based on a core-gene alignment built using 169 clade-specific core genes (see the "Methods" section). The red arrow indicates the Muc$^T$ type strain. **B** Within- and between-subspecies coregene nucleotide identities confirm the subspecies diversification defined on the phylogeny. **C** Per-host frequency of *A. muciniphila* subspecies assembled from metagenomes. All 174 mouse *A. muciniphila* genomes were reconstructed from stool metagenomes of laboratory mice

*muciniphila* strains). The subspecies were found to have similar within-subspecies genetic similarities—always exceeding 98% identity—while between-subspecies genetic similarities range from 95.4% genetic similarity between Amuc1 and Amuc4 to 98.6% between the more closely related Amuc2 and Amuc3 (Fig. 4B). This inter-subspecies genetic divergence was coupled also with a diversification of the functional profiles of the strains (Fig. 5A).

Amuc1 is the most prevalent subspecies in humans (47%), followed by Amuc2 and Amuc3 (27% and 24% respectively, Fig. 4C). To investigate whether these global prevalences were driven by particular host factors, we studied the distribution of *A. muciniphila* subspecies across host metadata (Additional file 2: Figure S10). In addition to a significantly higher prevalence of Amuc4 in non-Westernized human populations compared to non-Amuc4 (Fisher test *P*-value <0.001), we found that subspecies were differentially distributed across hosts. In particular, Amuc2 and Amuc3 are specific to humans and never found in mice and non-human primates, whereas Amuc1 and Amuc4 can be found in both humans and mice (Fig. 4C) but in different proportions, suggesting differential fitness of *A. muciniphila* subspecies in mice compared to humans. Notably, all *A. muciniphila* genomes we obtained from mice came from laboratory-held mice. Due to the lack of subspecies-specific marker genes, we were unable to extend prevalence analysis to samples lacking successfully reconstructed *Akkermansia* MAGs, but our data nonetheless suggests *Akkermansia* in mice may be acquired from humans and that there is a strong preference of laboratory mice to acquire only the Amuc1 (to which

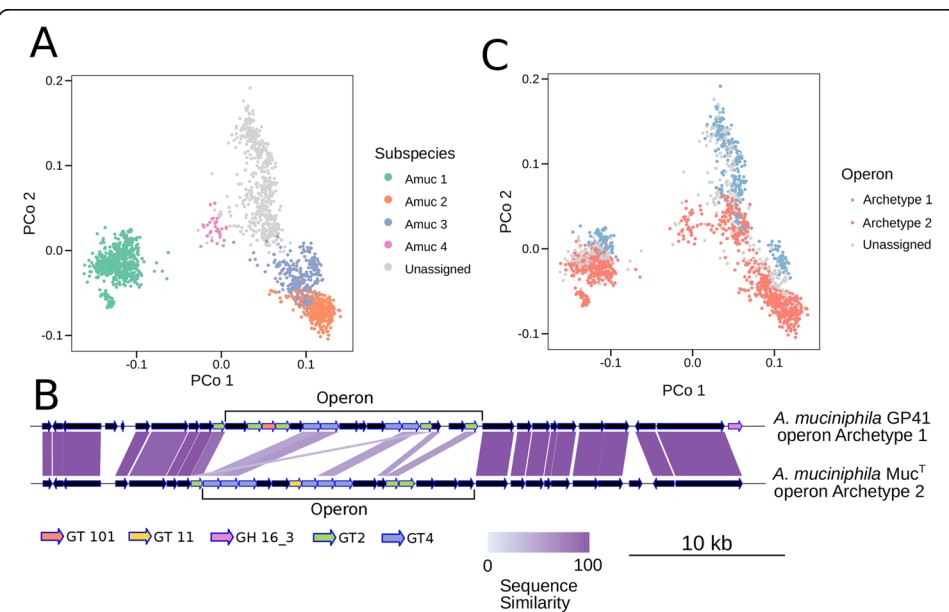

**Fig. 5** Functional diversification of *A. muciniphila* subspecies and cognate exopolysaccharide/LipidA synthesis operon. **A** Ordination analysis (Jaccard-distance-based PCoA using gene presence and absence information) reveals a diversification of gene repertoires of *A. muciniphila* subspecies. Genes found in less than 3% of strains were excluded. Subspecies designation is derived from the *A. muciniphila* phylogenetic tree in Fig. 4. **B** Operon archetypes putatively involved in exopolysaccharide/LipidA synthesis in *A. muciniphila* GP41 (operon archetype 1) and *A. muciniphila* Muc^T (operon archetype 2). **C** PCoA (same as in **A**) colored by operon archetype membership. Genomes in which neither operon could be found were labeled "Unassigned"

Muc$^T$ belongs) and Amuc4 *A. muciniphila* subspecies, which might have important implications for pre-clinical mice models.

### Two functionally related but distinct glycosyltransferase-rich operons are found in the *A. muciniphila* subspecies

Surface glycoconjugates are known to form a species- and sometimes even strain-specific glycan barcode, conferring bacteria with unique interaction properties [65]. Two well-conserved archetypes of a glycosyltransferase-rich operon were detected in the same genomic location in different *A. muciniphila* strains (Fig. 5B, Additional file 2: Figure S11). Both operon archetypes predominantly contain genes annotated as glycosyltransferases (GTs) belonging to two different CAZyme families (GT2 and GT4), albeit in different proportions: while archetype 1 contains five GT2 and four GT4 copies and archetype 2 contains three GT2 and six GT4 copies. GTs belonging to these families are typically involved in lipo- and/or exopolysaccharide biosynthesis, which are key in microbiota-host interactions [65]. However, despite both operon archetypes being mostly composed of functionally related GTs, only a few pairs of proteins displayed detectable but very remote sequence similarity (Fig. 5B). The two operon archetypes were notably differentially distributed among *A. muciniphila* subspecies: subspecies Amuc2 and Amuc4 always possessed archetype 2 (whenever detectable), whereas strains belonging to Amuc1 and Amuc3 had either archetype. *A. muciniphila* thus encodes one of two possibly very distantly related operons that are putatively involved in lipo/exopolysaccharide (LPS/EPS) biosynthesis functions, hinting at a possible divergence of their surface glycoconjugates as well as host-specific selective advantages.

### Discussion

The possibility of extracting whole (draft) microbial genomes of sufficient quality directly from metagenomic sequences [3, 9, 24–26, 66] together with the quickly increasing availability of metagenomes from diverse populations [67] is revolutionizing the way human-associated microbes can be studied and characterized [30–33]. Exploiting a combined set of over 18,600 metagenomic samples from multiple hosts, we studied the population genomics and genetic characteristics of bacterial strains belonging to the *Akkermansia* genus. While *A. muciniphila* is recognized as a keystone species of the human microbiome, current biomedical and translational research is still driven by the type strain Muc$^T$ [4], thus neglecting the genomic and phenotypic variability of conspecific strains as well as of closely related species. Previous comparative genomic efforts were able to survey only a fraction of the diversity in the *Akkermansia* genus we describe here due to the limited availability of isolate genomes [19, 20, 39]. At the same time, we extended similar ongoing work using MAGs for this genus [68] with a larger genome set and more diverse metagenomic sample set including non-human hosts, allowing us to explore aspects such as the association of *Akkermansia* abundances with phenotypes (particularly with respect to BMI), the in-depth analysis of some of its genetic features such as the machinery for vitamin B12 synthesis and a novel LPS/EPS operon, and the discovery of bacteriophages likely interacting with *Akkermansia* in the human gut.

Our analysis of 2420 *Akkermansia* genomes delineates four candidate species in addition to the well-defined *A. muciniphila* species. The five candidate species are prevalent in the human gut microbiome and are found in other mammals such as mice and non-human primates almost exclusively when living in man-made environments, suggesting that all *Akkermansia* candidate species are specifically adapted to the human gut. All candidate species have very high pairwise sequence similarity of the full-length 16S rRNA gene (> 98%) and substantially lower whole-genome similarity (< 90% for all pairs except SGB9223 and SGB9224). These unusual genomic characteristics are likely the reason why the diversity of the *Akkermansia* genus has been overlooked by extensive 16S rRNA gene amplicon sequencing surveys in the past. Most bacterial species at <95% genomic similarity have >3% divergence of the 16S rRNA gene [3], and the substantially different pattern observed in *Akkermansia* might suggest rapid genomic diversification of these clades in humans.

A potential instance of adaptive evolution in *Akkermansia* emerging from our analysis could be the loss of vitamin B12 synthesis capabilities that likely occurred independently in the ancestors of two candidate species. Vitamin B12 promotes symbiotic metabolic relationships between gut microbes [56]. Notably, a bidirectional syntrophy has been described between *A. muciniphila* Muc$^T$ and *Anaerobutyricum soehngenii* (formerly known as *Eubacterium hallii* [69]), with Muc$^T$ converting mucin into oligosaccharides and acetate which are used by the butyrate-producer *Anaerobutyricum soehngenii*, in turn providing (pseudo)vitamin B12 to enable propionate production by Muc$^T$ [70]. Hence, loss of vitamin B12 synthesis genes might have been metabolically favorable for *Akkermansia* candidate species SGB9223/9224 and *A. muciniphila* given the potential to syntrophically interact in this way with other species. Our results warrant future investigations also at the level of subspecies clades: for example, the presence of one of the two putative LPS/EPS operons we described in *A. muciniphila* may be driven by host-microbe interactions and host-specific factors such as diet or lifestyle.

Experimental efforts to investigate *Akkermansia*-host interactions that are currently fueled by findings of their potential role in biomedical settings (ranging from obesity [5–8] to cancer treatments [71, 72]) should consider some aspects of the genus-wide genomic diversity we are reporting here. For example, only *A. muciniphila* was confirmed in our analysis to be associated with decreased BMI and it is possible that the *A. muciniphila* subspecies might also display different strengths of association. Moreover, the limitations of animal-based experimental approaches should be particularly considered for *Akkermansia*: our finding that no *Akkermansia* candidate species is consistently detected in wild mice and primates may indeed suggest that these animals are not natural hosts for *Akkermansia,* and raises the question whether host-*Akkermansia* interactions can be meaningfully recapitulated in mice. Similarly, we obtained MAGs from only two out of four *A. muciniphila* subspecies from mice, suggesting that not all subspecies may be well adapted to the mouse gut, which has important implications for in vivo experiments. Further delineation of subspecies through bacterial isolation or single-cell sequencing will be required to answer this question conclusively.

The ecology of *Akkermansia* may however be driven not only by aspects of host fitness, as interaction with bacteriophages also potentially contribute to shaping the population structure and diversity of this microbe. While no known phages have been so far

linked with *Akkermansia* as a host, we identified at least four putative phages from gut viromes and gut metagenomes that display genomic regions recognized by spacer sequences in CRISPR-Cas loci in a species-specific manner. These putative phages also tend to co-occur with their cognate candidate species. Understanding the mechanisms of interaction between these phages and their targets could be an important experimental step in order to develop intervention strategies to modulate the presence and abundance of *Akkermansia* candidate species in the gut.

Our work provides a large-scale strain-level analysis of *Akkermansia* that can be the basis for future further investigations of this genus. We also further highlighted the potential of metagenomics-based investigations of bacteria of the human microbiome, which is particularly important given that most bacterial species have very little genomic information available from cultivation efforts. In our work, we also introduced new analysis types for MAG-based investigations complementing those already performed on other bacteria such as *Eubacterium rectale* [31], *Prevotella copri* [30], *Ruminococcus gnavus* [32], and *Faecalibacterium prausnitzii* [33]. Further extending and applying this approach to the hundreds of species in the human microbiomes will be crucial to better understand the bacterial constituents of human-associated microbial ecosystems.

## Material and methods

### Collection and taxonomic annotation of *Akkermansia* sp. genomes

The *Akkermansia* genomic sequences used in this work were retrieved from four sources: (i) newly sequenced *Akkermansia* genomes from cultivated strains [43], (ii) publicly available isolated genomes from NCBI (downloaded as of March 2020) that were labeled as *Akkermansia muciniphila* or *Akkermansia* sp., (iii) metagenome-assembled genomes (MAGs) coming from a collection of metagenomes from human microbiome by Pasolli et al. [3], and (iv) 166,518 additional MAGs reconstructed from 9172 metagenomes (Additional file 1: Table S4) obtained with a validated assembly-based pipeline similarly to Pasolli et al. [3].

For the 166,518 additional MAGs reconstructed specifically for this work, the metagenomes were assembled using metaSPAdes [73] if paired-end metagenomes were available, and MEGAHIT [74] otherwise. In both cases, default parameters were used. Contigs longer than 1500 nucleotides were binned into MAGs using MetaBAT2 [75]. We assigned MAGs to previously defined species-level genome bins (SGB) (Pasolli et al. [3]) based on whole-genome nucleotide similarity estimation using Mash [76] and only MAGs falling in the SGBs belonging to the *Akkermansiaceae* family were further considered. We then quality controlled the MAGs and genomes using checkM (version 1.1.3) [42] and kept genomes estimated to be high-quality according to genomic completeness >90% and genomic contamination <5%.

The above procedure resulted in a total of 2420 *Akkermansia* genomes being considered in this work (http://segatalab.cibio.unitn.it/data/Akkermansia_Karcher_et_al.html): 188 isolate genomes from NCBI (119 labeled as *Akkermansia muciniphila* and 69 labeled as *Akkermansia* sp.), 2226 MAGs, and 6 novel genomes coming from strains isolated from the human gut. The 2420 genomes were assigned to a total of five candidate species which includes the already recognized *Akkermansia muciniphila* species and

four additional SGBs: SGB9223, SGB9224, SGB9227, and SGB9228 as summarized in Table 1.

### Identification and comparison of the 16S rRNA genes from genomes and MAGs

16S rRNA genes were identified using Barrnap (version 0.9) with default parameters. We only considered extracted 16S rRNA gene sequences longer than 1000 nucleotides. We retained a total of 445 16S rRNA sequences (255 from isolate genomes and 190 from MAGs). Mapping all these sequences against the NCBI's 16S rRNA gene set identified 11 outlying 16S rRNA genes that had >98% whole-gene identity to a 16S rRNA gene of a family other than *Akkermansiaceae*, which we removed. We then aligned the sequences using mafft (version v7.471, [77]) with parameters: *--quiet --anysymbol --localpair --maxiterate 1000)* and computed pairwise edit distances between all sequences.

### Genome annotation and gene clustering

We detected and annotated ORFs on all genomes using Prokka (version 1.14) [78]. Coding sequences (CDS) were then assigned to a UniRef90 cluster [79] by performing a Diamond search (version 0.9.24) [80] of the CDS against the UniRef90 database (version 201906) and assigning a Uniref90-ID if the mean sequence identity to the centroid sequence is over 90% and if it covers more than 80% of the centroid sequence. Protein sequences that could not be assigned to any UniRef90 cluster were de novo clustered using MMseqs2 [81] following the Uniclust90 criteria [82].

### Whole-genome phylogenetic analysis

The phylogenetic analyses were performed with PhyloPhlAn3 [46], using either 400 universal marker genes when applied on the 2420 *Akkermansia* genomes or core genes when applied to each separate candidate species. Core genes of an *Akkermansia* candidate species were those ORFs whose assigned UniRef90 annotation (or de novo clustered gene family) was present in at least 80% of the genomes of the candidate species. The number of core genes varied across candidate species, with 1131 for SGB9223, 799 for SGB9224, 996 for SGB9228, and 169 for *A. muciniphila*. The phylogenies were obtained using PhyloPhlAn 3.0 using the following flags, in both cases, universal markers and specific core genes: "*--force_nucleotides --trim greedy --fast --diversity low*". The following tools with their specific parameter are used inside the PhyloPhlAn3 framework, diamond was used over blast to generate the database when the database sequences were in proteins:

**Table 1** Summary of the number of genomes per candidate species

|  | SGB9223 | SGB9224 | *A. muciniphila* | SGB9227 | SGB9228 | Total |
| --- | --- | --- | --- | --- | --- | --- |
| MAGs | 29 | 93 | 1802 | 4 | 298 | 2226 |
| Isolate genomes (NCBI) | 66 | 3 | 108 | 2 | 9 | 188 |
| Isolate genomes (generated) | 0 | 0 | 6 | 0 | 0 | 6 |
| Total | 95 | 96 | 1916 | 6 | 307 | 2420 |

- diamond (version v2.0.2.140, [80]) with parameters: *makedb* (to generate the database), *"blastx --quiet --threads 1 --outfmt 6 --more-sensitive --id 50 --max-hsps 35 -k 0"* (to map the dna) and *"blastp --quiet --threads 1 --outfmt 6 --more-sensitive --id 50 --max-hsps 35 -k 0 "*
- blast (version 2.10.1+, [83, 84]) with parameters: *"makeblastdb -parse_seqids -dbtype nucl* and *blastn -outmft 6 -max_target_seqs 1000000"*
- Mafft (version v7.471, [77]) with parameters : *"--quiet --anysymbol --localpair --maxiterate 1000"*
- trimal (v1.4.rev15 build[2013-12-17], [85]) with parameters: *":-gappyout"*
- RAxML (version 8.2.12, [86]) with parameters: *"-p 1989 -m GTRCAT -x 1989 -# 100 -f a"*

## Relative abundance estimation of candidate species

In order to estimate the presence and relative abundance of the *Akkermansia* candidate species, we extended the database of unique marker genes of MetaPhlAn 3.0 [49, 87] with those of the newly defined *Akkermansia* candidate species. Unique marker genes were defined starting from the core genes of each of the 5 *Akkermansia* candidate species identified on the clustered gene families described above. Core genes of each *Akkermansia* candidate species were divided into 150 nucleotide fragments and then aligned against the genomes of all SGBs including both the other *Akkermansia* candidate species as well as the whole set of bacterial and archaeal SGBs defined in Pasolli et al. [3] using bowtie2 (version 2.3.5.1; --sensitive option) [88]. A core gene was considered present in a genome if at least one of the gene's fragments was mapping against it. Core genes never found in more than 1% of the sequences included in any other SGBs were selected as marker genes, obtaining 39, 22, 115, 100, and 135 species-specific unique markers for SGB9223, SGB9224, SGB9227, SGB9228, and *A. muciniphila*, respectively. MetaPhlan 3 was then used with default parameters. The prevalence of candidate species was defined as the percentage of samples in which the candidate species was detected. Similarly, the prevalence of the *Akkermansia* genus was defined as the percentage of samples in which at least one of the candidate species could be detected.

## Covariation among candidate species

Covariation among relative abundances of *Akkermansia* candidate species was assessed in 4171 human metagenome samples in which at least one of the candidate species was detected (out of the 11,014 metagenomes from humans, Additional file 1: Table S2) by performing pairwise Spearman's correlations (cor.test in the stats R package [89]). We corrected for multiple testing using the Benjamini-Hochberg procedure at 10% FDR.

## Association between candidate species and metadata parameters

The association between relative abundances of *Akkermansia* candidate species and host BMI, age, and gender was analyzed in 3311 human metagenomic samples from 22 datasets in which this information was available (Additional file 1: Table S3). For continuous variables (age and BMI), Spearman's correlations were computed using the *pcor.test* function from the *ppcor* R package [90] controlling for the remaining

covariates. Resulting correlations were used as input in the *metacor* function from the *meta* R package [91] using Fisher's Z transformation of correlations and the Paule-Mandel estimator of between-study variance in the random effects model. For categorical variables (sex), an ordinary least squares (OLS) model was first used to adjust for age and BMI. Statistical significance (Wald test) and effect sizes (standardized mean difference) of the associations were extracted from the sex beta coefficients. Resulting effect sizes were inverse-variance averaged using the Paule-Mandel estimator of between-study variance as implemented in the statsmodels python library [92] and custom code. We corrected for multiple testing using the Benjamini-Hochberg procedure at 10% FDR.

### Identification of corrin ring biosynthesis genes

Anaerobic corrin ring biosynthesis gene names were obtained from [93]. Corresponding KEGG Orthologs (KOs) were then identified in the clustered gene sequences (see above) using KOFAM [94]. Only those hits that passed the optimized bit-score cutoffs from KOFAM were considered. We found a total of 316 genomes with at least one significant hit.

### Determination of vitamin B12 utilization and production by *A. muciniphila* and *A. glycaniphila*

The type strains *A. muciniphila* Muc$^T$ (ATCC BAA-835) and *A. glycaniphila* Pyt$^T$ (DSM 100705) were grown in minimal bicarbonate buffered medium supplemented with 0.6% threonine, 30 mM 3:1 Glc:GlcNAc, and a vitamin mixture with and without added vitamin B12 [95]. Cultures were inoculated with a preculture produced on mucin-supplemented medium. At several time points (0, 3, 8.5, 21, 28, 33, 48 h), a 1-mL sample was collected to measure cell density (OD 600nm) and determine propionate concentration as a proxy for vitamin B12 production. Substrate utilization and metabolite production were quantified on a Thermo Scientific HPLC system equipped with an Agilent Metacarb 67H 300 × 6.5 mm column. The column was kept at 45°C, running 0.005 M $H_2SO_4$ eluent at a flow rate of 1 mL/min. Detection was performed using a refractive index detector. All measurements were performed in duplicate.

### LPS operon identification

The pangenome of *A. muciniphila* was reconstructed using the UniRef90 assignments and complemented with the de novo clustered gene families (see above). Pan-genes were then also annotated with CAZy using a local dbCAN distribution [96] (database version V9 with suggested E-value and HMM coverage cutoffs of 1E−18 and 0.35, respectively). We specifically focused on the differential copy number and distribution of the glycosyltransferase enzymes class 2 and 4 (GT2/GT4) in the *A. muciniphila* genomes. We observed two groups within this set of genes that were co-present and mutually exclusive in genomes, suggesting a large structural variation and operon-type distribution of genes. We then determined the two putative archetypes by manual inspection of gene distribution and order on isolate genomes. Finally, the detected dichotomy was confirmed by performing BLAST on operon genes (including bordering

genes from the isolate genomes) against all genomes and observing their presence/absence (Additional file 2: Figure S11).

### CAZy annotation and gene clustering

dbCAN2 ([97], database version 07312020) was used to annotate centroid sequences of gene clusters (see above) with Carbohydrate-Active enZYmes (CAZY) information [98]. dbCAN2 was used with default parameters, and hits with an E-value >10E−15 and those that covered less than 35% of a given dbCAN2-HMM were removed.

### Retrieval of CRISPR spacers in viruses from metagenomes and viromes

Metagenomes enriched for virus-like particles (i.e., viromes) were retrieved through SRA [99] from 708 samples from five studies [100–104]. Samples were uniformly preprocessed with TrimGalore version 0.4.4 [105] to remove low quality and short reads (Phred quality <20, read length < 75; parameters: --stringency 5 --length 75 --quality 20 --max_n 2 --trim-n). Reads aligning to the human genome (hg19) were identified and subsequently removed via mapping with Bowtie2 version 2.4.1 [88] in global mode. Raw reads were assembled with metaSPAdes [73] version 3.10.1 (k-mer sizes: -k 21,33, 55,77,99,127). The efficacy of viral enrichment of each virome was evaluated with ViromeQC [106], and 126 out of 708 samples had an enrichment higher than 50-fold. Contigs (a) longer than 1500bp; (b) originating from highly enriched viromes (i.e., enrichment ≥ 50x); (c) found binned in the same Species-level Genome Bin [3] in less than 30 metagenomes; and (d) found in the unbinned fraction of more than 20 metagenomes [3] were retained as putative viral contigs. After this, contigs originating from non-highly enriched viromes with a high sequence similarity were added to the collection (BLAST identity ≥ 80%, length ≥ 1000 nucleotides, by using BLAST, version 2.6.0 [107]). Sequences homologous to the virome-derived contigs were searched in unbinned contigs of Pasolli et al. with mash version 2.0 [76], and contigs with a distance lower than 10% (*p*-value ≤ 0.05) to any viral contig were added to the collection. Finally, we added 699 full genomes of taxonomically annotated gut bacteriophages from RefSeq, release 99 [63] that were also found in at least 20 metagenomes of the unbinned fraction of Pasolli et al. [3].

Putative viral contigs were then clustered at 70% identity with VSearch version 2.14.2 [108] (parameters --cluster_fast --id 0.7 --strand both) and further grouped if they shared more than one third of their sequence at 90% sequence identity or more to produce 1345 "Viral Clusters" (VCs) that were further analyzed.

CRISPR arrays and Cas genes were predicted using CRISPRCasTyper version 1.2.1 (default parameters) [62]. In order to understand potential interaction of candidate species and VCs, we aligned spacer sequences against VCs with BLAST version 2.2.31 (parameters -task blastn-short -gapopen 1 -gapextend 2 -penalty -1 -reward 1 -evalue 1 -word_size 10). Near-perfect matches were defined as matches with an edit distance ≤ 2. CRISPR-Cas loci structures were plotted using DNA Features Viewer version 3.0.3 [109]. Sequence logos were generated using Logomaker version 0.8 [110]. We used spacers from orphan as well as non-orphan CRISPR arrays for all spacer-based analyses (Fig. 3D, F–H). For subsequent analysis, we considered only those four VCs where at

least 5% of the genomes of a given candidate species had at least one spacer sequence with a hit.

In order to detect the presence of a VC in a metagenome, we mapped a total of 13, 381 gut metagenomes against VC contigs with Bowtie2 [88] version 2.4.1 in global mode. Breadth and depth of coverage were evaluated for each VC with bedtools version 2.29.1 [111] (genomecov command, default parameters). Only alignments with a Bowtie2 alignment score (AS:i tag) greater than −50 were considered. A VC was considered detected if at least one sequence in the cluster had a breadth of coverage of at least 50%. Differential abundance of VCs in subspecies was assessed with two-sided Wilcoxon rank-sum tests. *P*-values of < 0.01 were considered significant.

## Supplementary Information

---

**Additional file 1:** Supplementary Tables S1 to S4.

**Additional file 2.** Supplementary Figures S1 to S11.

**Additional file 3.** Review history.

---

**Peer review information**

**Review history**
The review history is available as Additional file 3.

**Authors' contributions**
NS, WdV, MVC, and NK conceived and supervised the study. NK, EN, MP, ABM, PM, MZ, FC, and MVC performed the data acquisition. NK, EN, MP, ABM, MC, PM, MZ, FC, DG, SM, and MVC performed the data analysis. TPNB, HLPT, and WdV designed and performed the in vitro experiments. NK, EN, MC, AC, MA, MVC, WdV, and NS performed the data interpretation and wrote the manuscript. All authors read and approved the final manuscript.

**Funding**
This work was supported by the European Research Council (ERC-STG project MetaPG-716575) to NS, by MIUR 'Futuro in Ricerca' (grant no. RBFR13EWWI_001) to NS, by the European H2020 program (ONCOBIOME-825410 project and MASTER-818368 project) to NS, by the National Cancer Institute of the National Institutes of Health (1U01CA230551) to NS, and by the Premio Internazionale Lombardia e Ricerca 2019 to NS. This work was partly supported by the SIAM Gravitation Grant 024.002.002 and the 2008 Spinoza Award of the Netherlands Organization for Scientific Research to WMdV. We thank the Institute of Biotechnology, University of Helsinki (Finland), for providing both Illumina and PacBio sequences of new isolates.

**Availability of data and materials**
The 2420 *Akkermansia* genomes and MAGs considered in this work are available at http://segatalab.cibio.unitn.it/data/Akkermansia_Karcher_et_al.html as well as in Zenodo under the following accession 0.5281/zenodo.5018705 [112].

## Declarations

**Ethics approval and consent to participate**
Not applicable.

**Consent for publication**
Not applicable

**Competing interests**
WMdV is co-founder and holds stock in A-mansia Biotech Belgium. All the other authors declare that they have no competing interests.

**Author details**
[1]Department CIBIO, University of Trento, Trento, Italy. [2]Novo Nordisk Foundation Center for Basic Metabolic Research, Faculty of Health and Medical Sciences, University of Copenhagen, Copenhagen, Denmark. [3]Laboratory of Microbiology, Wageningen University, Wageningen, The Netherlands. [4]Current address: Nestlé Institute of Health Sciences, Nestlé Research, Société des Produits Nestlé S.A., Lausanne, Switzerland. [5]Human Microbiome Research Program, Faculty of Medicine, University of Helsinki, Helsinki, Finland. [6]IEO, European Institute of Oncology IRCCS, Milan, Italy.

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

## 

