## [**Additional file 3.** Review history. · Genome Biology]

Review History

First round of review

Reviewer 1

Are you able to assess all statistics in the manuscript, including the appropriateness of statistical tests used? Yes, and I assessed the statistics in my report.

Comments to author:

The authors report a comprehensive and thoughtful analysis of the genus *Akkermansia*, which has frequently been associated with host health parameters. More studies like these are required for the development of effective probiotic therapies. I have a few requests for clarifications of some of the figures and text.

1. Page 4, Figure 1:

* I realize that you define the acronym SGB in the text, but can you please also define it in the legend?

* The scale that you are using in panel B makes it difficult to visualize your points made in the text. The boxplots are smashed into the upper part of the plots. I commend you for using the same scale for all of them, but my eyesight was failing - it's a trade-off. Could you consider changing the scale of the y-axis?

2. Page 6, Figure 2, panel A:

* It is difficult to see the boundaries between the categories on the x-axis. I think that there are faint grey lines between them, but they are so light that I may be hallucinating. Because the prevalence of some of the SGBs is so low, it's unclear where Adults, Children, and Humanized mouse categories begin and end.

* The y-axis is labeled prevalence (%), and I read your methods section labeled: Relative abundance estimation of candidate species. What is the number plotted in the bar graph? Is it the mean relative abundance of that candidate species in the host, e.g. mean relative abundance of SGB9228 in all of the wild mouse samples? If so, which genetic marker(s) is it based on? Or is it absence/presence - the % of hosts in which the candidate species was present?

3. Page 7, lines 198-200:

Do you know whether the strains being used as supplements are carrying the resistance genes? Are the genomes of the strains used in previous supplementation studies available for analysis? Your arguments would be stronger regarding potential threats if analysis of strains used in supplements revealed antibiotic resistance genes.

4. Page 8, lines 227-232

By identifying corrin ring biosynthesis genes as a proxy for vitamin B12 synthesis capability [54], we confirm that candidate species SGB9227 and SGB9228 largely encode the proteins necessary to produce vitamin B12 (75% of SGB9227 and 92% of SGB9228 MAGs missing the *cblF/cblA* gene, 25% and 86% respectively carrying all corrin ring biosynthesis genes), while those genes were never found in *A. muciniphila*, SGB9223, nor SGB9224 (Fig. 2E).

This statement is not supporting my observations of the figure - or I am misunderstanding

something. I could see where 75% of SGB9227 MAGS are missing the *cbiA* gene based on Figure 2E. However, is Figure 2E showing that 92% of SGB9228 MAGS are missing the *cbiA* or *cbiF* genes? All of the bars representing the genes appear to be above 0.5 so far as I can tell. It may be easier to see if the y-axis were stretched out a bit.

5. Page 9, Figure 3

* In 3A, where does the category "Other" appear? Is it present in any of the candidate species? If so, it is not visible to the naked eye.

* In 3G, it would be helpful if the order of the SGBs in the key was the same as the order of the panels shown for VC M689. I think that the first panel shown is *A. muciniphila*.

6. Abstract

These candidate species appeared to be human-specific as they were detected in mice and non-human primates almost exclusively when kept in captivity.

Page 11, lines 332-335

Notably, all *A. muciniphila* genomes we obtained from mice came from laboratory-held mice. Our data thus suggest *Akkermansia* in mice may be acquired from humans and that there is a strong preference of laboratory-mice to acquire only the Amuc1 and Amuc4 *A. muciniphila* subspecies.

I'm not convinced that the species are human-specific. If Figure 2A is based on presence/absence, I am wondering whether there is a sampling issue. Understandably, you have far more samples from humans and captive mice than the other categories. Furthermore, I would argue that the presence of the species is strongly diet-dependent and that the diets of all of those critters in the wild versus humans and captive critters are affecting presence of *Akkermansia* species in the gut.

Also, in the case of laboratory mice, if *Akkermansia* is acquired from humans, I would expect Figure 4C to show that the subspecies Amuc 2 and Amuc 3 are at a higher proportion in laboratory mice than Amuc 4.

I'm not sure that the statement that these candidate species are human-specific will withstand the test of time in the event that many, many more wild hosts are added to the databases - but that's how science works. Report the exciting findings and inspire future work. I concede that this is a minor point.

Reviewer 2

Are you able to assess all statistics in the manuscript, including the appropriateness of statistical tests used? No, I do not feel adequately qualified to assess the statistics.

Comments to author:

Here, Karcher and colleagues set out to provide a large-scale population genomic analysis of the *Akkermansia* genus, representing 2420 genomes across diverse host populations with valuable metadata. Their comprehensive genomic characterization of *Akkermansia* species revealed 5 distinct species within this genus, associated with the mammalian host.

Interestingly they demonstrate marked specificity of *Akkermansia* sp. colonizing the human gut and note that colonization of mice and human primates appear to be largely a consequence of human-made environments.

They also provide striking evidence of *Akkermansia* sp. co-exclusion, whereby they rarely found more than one sp. co-occurring within a host. Considering the potential benefit of *A.*

mucinophila as a probiotic for a range of metabolic diseases, the authors delve deeper into the genetics and functional characteristics of this species.

They demonstrate that there are at least 4 subspecies within the *A. muciniphila* sp. and notably that only subsp *A.muc 1 + 4* are detected in mice.

I have some minor comments:

The authors note that 'experimental *Akkermansia* research still heavily relies on the type strain *Akkermansia muciniphila* MucT (ATCC BAA-835)' and it would be valuable for the authors to identify what subspecies this type strain falls into. The primary importance of this is that preclinical mouse models are often used to assess efficacy of therapeutics of strains as probiotics, and a challenge will be that *A.muc 2 + 3* do not appear to colonize mice. Given that the type strain has been used to date it likely falls into either *A.muc1* or *4*, but worth confirming and noting the potential for subspecies selection as a confounding factor in pre-clinical models. Additionally, did the authors detect *A.muc 2 + 3* in NHP? (in consideration of suitable preclinical models for probiotics studies for such subspecies, lack of murine colonization should not negate these subsp. potential probiotic use in human metabolic disorders.

Given the authors finding that there are 4 different subsp. within *A.muciniphila*, are each of these subsp. independently inversely correlated with BMI ? (similar to the finding with *A.muciniphila* at the species level).

The relevance here is that there may be a bias to study *A.muc 1 + 4* based on their colonization of murine models, and it would be important to note if these subsp. are truly inversely correlated with BMI, if they are to be considered to be a potential probiotic

The authors nicely demonstrate that there is significant co-exclusion of *Akk* sp. within an individual host, it would be interesting to extend this to the subsp level of *A.muciniphila* and determine if multiple *A.muc* subsp. can co-occur within an individual host.

Overall, I found this manuscript to be very well written, experiments and analysis appear to be meticulous and the findings represent a significant advance over previously published studies, and therefore will be of high value to the field.

Reviewer #1: The authors report a comprehensive and thoughtful analysis of the genus *Akkermansia*, which has frequently been associated with host health parameters. More studies like these are required for the development of effective probiotic therapies.

We thank the Reviewer for the positive feedback. We agree studies on the genomic diversity of key taxa are necessary for the development of microbiota-modulation strategies.

I have a few requests for clarifications of some of the figures and text.

1. Page 4, Figure 1:

* I realize that you define the acronym SGB in the text, but can you please also define it in the legend?

We thank the Reviewer for this pertinent suggestion. The SGB acronym is now defined in the legend as well as in the main text.

* The scale that you are using in panel B makes it difficult to visualize your points made in the text. The boxplots are smashed into the upper part of the plots. I commend you for using the same scale for all of them, but my eyesight was failing - it's a trade-off. Could you consider changing the scale of the y-axis?

In order to make this panel more readable and informative, we opted to stretch the y-axis resolution and remove the sub-panel for *A. glycaniphila*, making it possible to better see the details of the boxplots (see below).

2. Page 6, Figure 2, panel A:

* It is difficult to see the boundaries between the categories on the x-axis. I think that there are faint grey lines between them, but they are so light that I may be hallucinating. Because the prevalence of some of the SGBs is so low, it's unclear where Adults, Children, and Humanized mouse categories begin and end.

We thank the Reviewer for pointing this out. We now added more visible lines between different species and different categories to make the separation more clear:

* The y-axis is labeled prevalence (%), and I read your methods section labeled: Relative abundance estimation of candidate species. What is the number plotted in the bar graph? Is it the mean relative abundance of that candidate species in the host, e.g. mean relative abundance of SGB9228 in all of the wild mouse samples? If so, which genetic marker(s) is it based on? Or is it absence/presence - the % of hosts in which the candidate species was present?

We apologize for the lack of clarity. As the Reviewer correctly guessed, "prevalence (%)" in the y-axis corresponds to the % of samples in which the candidate species or the genus was detected. To make this more clear, we now explain how prevalence was defined in the methods:

"Prevalence of candidate species was defined as the percentage of samples in which the candidate species was detected. Similarly, the prevalence of the *Akkermansia* genus was defined as the percentage of samples in which at least one of the candidate species could be detected."

3. Page 7, lines 198-200:

Do you know whether the strains being used as supplements are carrying the resistance genes? Are the genomes of the strains used in previous supplementation studies available for analysis? Your arguments would be stronger regarding potential threats if analysis of strains used in supplements revealed antibiotic resistance genes.

We thank the reviewer for the interest. The *Akkermansia* strains used in previous intervention trials were either the type strains or strains whose genome/nature have not been disclosed. Accordingly, our remark was a general warning for the scientific community and regulatory bodies. We have now rephrased this in the following way and referred to published safety studies on *A. muciniphila* with a specific additional focus on the type strain:

"We queried all genomes for the presence of this plasmid-derived sequence and found 55 genomes (2.2% overall prevalence) in which we could detect at least 50% of the sequence of RSF1010 at 70% average sequence identity or higher. A total of 49 of the 55 instances were found in *A. muciniphila* (2.5% prevalence in *A. muciniphila*). In all 55 positive cases, these genes were found on contigs larger than the plasmid (~8 kb), suggesting that they may be integrated into the bacterial genome (as also reported in [19]). Of note, the *A. muciniphila* type strain MucT carries no antibiotic resistance genes and its use does not raise any antibiotic resistance concern as also indirectly confirmed by dose scaling pilot studies in humans and toxicological studies in rabbit and other model organisms [8,54]; however, ongoing and future human trials with strains different from the type strain should carefully consider their antibiotic resistance potential. In conclusion, although the rare occurrence of antibiotic resistance genes from plasmid RSF1010 in some *A. muciniphila* genomes has evident safety implications for their use in therapeutic applications, our findings indicate that *Akkermansia* candidate species mostly lack genetic means to defend themselves against currently used antibiotics."

4. Page 8, lines 227-232

By identifying corrin ring biosynthesis genes as a proxy for vitamin B12 synthesis capability [54], we confirm that candidate species SGB9227 and SGB9228 largely encode the proteins necessary to produce vitamin B12 (75% of SGB9227 and 92% of SGB9228 MAGs missing the *cbiF/cbiA* gene, 25% and 86% respectively carrying all corrin ring biosynthesis genes), while those genes were never found in *A. muciniphila*, SGB9223, nor SGB9224 (Fig. 2E).

This statement is not supporting my observations of the figure - or I am misunderstanding something. I could see where 75% of SGB9227 MAGS are missing the *cbiA* gene based on Figure 2E. However, is Figure 2E showing that 92% of SGB9228 MAGS are missing the *cbiA* or *cbiF* genes? All of the bars representing the genes appear to be above 0.5 so far as I can tell. It may be easier to see if the y-axis were stretched out a bit.

We realize that we didn't express ourselves clearly enough, and that the referenced paragraph can be misunderstood. We have rephrased the entire paragraph to now describe the data/Figure 2E unambiguously:

“By identifying corrin ring biosynthesis genes as a proxy for vitamin B12 synthesis capability [55], we confirm that the large majority of MAGS from candidate species SGB9227 and SGB9228 encode most proteins involved in producing vitamin B12 (75% of SGB9227 MAGs encode all proteins except *CbiA*; 92% of SGB9228 MAGs encode all proteins except *CbiF*), while those genes were never found in *A. muciniphila*, SGB9223, nor SGB9224 (Fig. 2E)”

5. Page 9, Figure 3

* In 3A, where does the category "Other" appear? Is it present in any of the candidate species? If so, it is not visible to the naked eye.

The 'Other' category is very rare, and indeed extremely difficult to see. We have removed this category from the legend and from the plot to avoid confusion.

* In 3G, it would be helpful if the order of the SGBs in the key was the same as the order of the panels shown for VC M689. I think that the first panel shown is *A. muciniphila*.

We have reordered the plots to be in line with the order of the legend.

6. Abstract

These candidate species appeared to be human-specific as they were detected in mice and non-human primates almost exclusively when kept in captivity.

Notably, all *A. muciniphila* genomes we obtained from mice came from laboratory-held mice. Our data thus suggest *Akkermansia* in mice may be acquired from humans and that there is a strong preference of laboratory-mice to acquire only the Amuc1 and Amuc4 *A. muciniphila* subspecies.

I'm not convinced that the species are human-specific. If Figure 2A is based on presence/absence, I am wondering whether there is a sampling issue. Understandably, you have far more samples from humans and captive mice than the other categories. Furthermore, I would argue that the presence of the species is strongly diet-dependent and that the diets of all of those critters in the wild versus humans and captive critters are affecting presence of *Akkermansia* species in the gut.

Also, in the case of laboratory mice, if *Akkermansia* is acquired from humans, I would expect Figure 4C to show that the subspecies Amuc 2 and Amuc 3 are at a higher proportion in laboratory mice than Amuc 4.

We appreciate this comment and largely agree with it (see below). Specifically regarding the distribution of the different *Akkermansia muciniphila* subspecies, we think that mice and humans having differential subspecies proportions isn't at odds with the hypothesis that mice predominantly acquire their strains from humans: some subspecies might be more suited than others for the mouse gut environment, which would select for some subspecies. We have added a remark in the main text to make our hypothesis more clear:

“In particular, Amuc2 and Amuc3 are specific to humans and never found in mice and non-human primates, whereas Amuc1 and Amuc4 can be found in both humans and mice (**Fig. 4C**) but in different proportions, suggesting differential fitness of *A. muciniphila* subspecies in mice compared to humans.”

I'm not sure that the statement that these candidate species are human-specific will withstand the test of time in the event that many, many more wild hosts are added to the databases - but that's how science works. Report the exciting findings and inspire future work. I concede that this is a minor point.

Overall, we agree with the reviewer that diet might influence the prevalence pattern of *Akkermansia* species, and that the sampling of new hosts might challenge this conclusion in the future. We have added a remark about the potential bias of diet and sampling in the concluding paragraph of the main text:

“Despite potential biases due to uneven sampling and effects of diet, these data suggest a marked specificity of *Akkermansia* candidate species for the human gut (with the exception of *A. glycaniphila*), and while strains from these candidate species can colonize mice and non-human primates, such colonization appears to be predominantly a consequence of man-made environments, suggesting colonization from care-taking humans as a plausible mechanism.”

We furthermore accordingly changed the title of the manuscript to “Genomic diversity and ecology of human-associated *Akkermansia* species in the gut microbiome revealed by extensive metagenomic assembly”

Reviewer #2: ===

Here, Karcher and colleagues set out to provide a large-scale population genomic analysis of the *Akkermansia* genus, representing 2420 genomes across diverse host populations with valuable metadata. Their comprehensive genomic characterization of *Akkermansia* species revealed 5 distinct species within this genus, associated with the mammalian host.

Interestingly they demonstrate marked specificity of *Akkermansia* sp. colonizing the human gut and note that colonization of mice and human primates appear to be largely a consequence of human-made environments.

They also provide striking evidence of *Akkermansia* sp. co-exclusion, whereby they rarely found more than one sp. co-occurring within a host. Considering the potential benefit of *A. muciniphila* as a probiotic for a range of metabolic diseases, the authors delve deeper into the genetics and functional characteristics of this species.

They demonstrate that there are at least 4 subspecies within the *A. muciniphila* sp. and notably that only subsp *A.muc 1 + 4* are detected in mice.

We thank the Reviewer for their summary of our work and for the constructive feedback.

I have some minor comments:

The authors note that 'experimental *Akkermansia* research still heavily relies on the type strain *Akkermansia muciniphila* MucT (ATCC BAA-835)' and it would be valuable for the authors to identify what subspecies this type strain falls into. The primary importance of this is that preclinical mouse models are often used to assess efficacy of therapeutics of strains as probiotics, and a challenge will be that *A.muc 2 + 3* do not appear to colonize mice. Given that the type strain has been used to date it likely falls into either *A.muc1* or *4*, but worth confirming and noting the potential for subspecies selection as a confounding factor in pre-clinical models.

We thank the Reviewer for raising this interesting point. We found MucT to fall into subspecies *Amuc 1*. We now added this information in the main text as well as its implications for pre-clinical research:

“Due to the lack of subspecies-specific marker genes we were unable to extend prevalence analysis to samples lacking successfully reconstructed *Akkermansia* MAGs, but our data nonetheless suggests *Akkermansia* in mice may be acquired from humans and that there is a strong preference of laboratory mice to acquire only the *Amuc1* (to

which MucT belongs) and Amuc4 *A. muciniphila* subspecies, which might have important implications for pre-clinical mice models.”

Additionally, did the authors detect A.muc 2 + 3 in NHP? (in consideration of suitable preclinical models for probiotics studies for such subspecies, lack of murine colonization should not negate these subsp. potential probiotic use in human metabolic disorders.

We unfortunately obtained merely 3 *Akkermansia muciniphila* MAGs from NHPs, which is not enough data to make any claim about subspecies enrichment within NHPs. As discussed below, it is too challenging to assign *Akkermansia muciniphila* strains to its subspecies Amuc1-4 without relying on reconstructed MAGs. The absence of Amuc 2 and Amuc 3 in NHP is thus what our data shows but cannot be generalized given the low number of available MAGs from NHPs. Therefore, we prefer not to make claims regarding their pre-clinical potential.

Given the authors finding that there are 4 different subsp. within *A.muciniphila*, are each of these subsp. independently inversely correlated with BMI ? (similar to the finding with *A.muciniphila* at the species level).

The relevance here is that there may be a bias to study *A.muc* 1 + 4 based on their colonization of murine models, and it would be important to note if these subsp. are truly inversely correlated with BMI, if they are to be considered to be a potential probiotic

The authors nicely demonstrate that there is significant co-exclusion of *Akk* sp. within an individual host, it would be interesting to extend this to the subsp level of *A.muciniphila* and determine if multiple *A.muc* subsp. can co-occur within an individual host.

We thank the Reviewer for raising these two very interesting questions. However, intra-species (i.e. intra-SGB) sub-species assignments are only currently possible based on whole MAGs, and obtaining multiple MAGs from the same SGB in the same sample is extremely rare since very similar strains tend to co-assemble and be consequently excluded by the QC steps (see Pasolli et al, Cell, 2019). While we tried to derive presence/absence calls as well as relative abundance estimates for each *Akkermansia muciniphila* subspecies using the very few subspecies-specific marker genes or the few SNV that are subspecies-specific, our benchmarks highlighted the procedure was too noisy to be reliable and robust. So while we agree that the question is very interesting, we unfortunately had to conclude that it cannot be addressed without conducting cultivation or single-cell sequencing experiments. We now add a comment in the discussion about the relevance of trying to answer this question in future studies:

“Moreover, the limitations of animal-based experimental approaches should be particularly considered for *Akkermansia*: our finding that no *Akkermansia* candidate species is consistently detected in wild mice and primates may indeed suggest that these animals are not natural hosts for *Akkermansia* and raises the question whether host-*Akkermansia* interactions can be meaningfully recapitulated in mice. Similarly, we obtained MAGs from only two out of four *A. muciniphila* subspecies from mice, suggesting that not all subspecies may be well adapted to the mice gut, which has

important implications for *in vivo* experiments. Further delineation of subspecies through bacterial isolation or single-cell sequencing will be required to answer this question conclusively.”

Overall, I found this manuscript to be very well written, experiments and analysis appear to be meticulous and the findings represent a significant advance over previously published studies, and therefore will be of high value to the field.

We again thank the Reviewer for this encouraging assessment of our work.